# Non-Melanoma Skin Cancer Clearance after Medical Treatment Detected with Noninvasive Skin Imaging: A Systematic Review and Meta-Analysis

**DOI:** 10.3390/cancers14122836

**Published:** 2022-06-08

**Authors:** Stefania Guida, Antonio Alma, Kaleci Shaniko, Johanna Chester, Silvana Ciardo, Ilaria Proietti, Roberta Giuffrida, Iris Zalaudek, Marco Manfredini, Caterina Longo, Francesca Farnetani, Giovanni Pellacani

**Affiliations:** 1Dermatology Unit, Department of Surgical, Medical, Dental and Morphological Sciences Related to Transplant, Oncology and Regenerative Medicine, University of Modena and Reggio Emilia, 41124 Modena, Italy; antonioalma@virgilio.it (A.A.); shaniko.kaleci@unimore.it (K.S.); johannamary.chester@unimore.it (J.C.); ciardo.silvana@aou.mo.it (S.C.); marco.manfredini@unimore.it (M.M.); caterina.longo@unimore.it (C.L.); francesca.farnetani@unimore.it (F.F.); 2Dermatology Unit “Daniele Innocenzi”, Department of Medical-Surgical Sciences and Bio-Technologies, Sapienza University of Rome, Fiorini Hospital, Polo Pontino, 04019 Terracina, Italy; ilaria.proietti@uniroma1.it; 3Department of Clinical and Experimental Medicine, Section of Dermatology, University of Messina, 98122 Messina, Italy; robgiuffrida@unime.it; 4Dermatology Clinic, Hospital Maggiore of Trieste, University of Trieste, 34127 Trieste, Italy; izalaudek@units.it; 5Centro Oncologico ad Alta Tecnologia Diagnostica, Azienda Unità Sanitaria Locale-Istituto di Ricovero e Cura a Carattere Scientifico di Reggio Emilia, 42122 Reggio Emilia, Italy; 6Dermatologic Unit, Department of Clinical Internal, Anesthesiological and Cardiovascular Sciences, Sapienza University of Rome, 00185 Rome, Italy; giovanni.pellacani@uniroma1.it

**Keywords:** basal cell carcinoma, squamous cell carcinoma, non-melanoma skin cancer, nonsurgical treatment, dermoscopy, reflectance confocal microscopy, optical coherence tomography

## Abstract

**Simple Summary:**

Non-melanoma skin cancers (NMSC) represent about one-third of all malignancies. While surgery is the current gold standard treatment, many nonsurgical approaches are available for selected cases. Currently, there are no studies concerning the overall impact of dermoscopy, optical coherence tomography (OCT) and reflectance confocal microscopy (RCM) for NMSC treatment monitoring. Therefore, we aim to review the current literature and provide an updated summary of noninvasive skin imaging in NMSC medical treatment management and the diagnostic accuracy of the most advanced technologies.

**Abstract:**

Background/Objectives: Non-melanoma skin cancer (NMSC) treated with nonsurgical therapies can be monitored with noninvasive skin imaging. The precision of dermoscopy, reflectance confocal microscopy (RCM) and optical coherence tomography (OCT) in detecting clearance is unclear. We aim to report the proportion of persisting tumors identified with noninvasive technologies available in the literature. Methods: A systematic literature search was conducted on the PubMed and Cochrane Public Library Databases for articles published prior to November 2021. Statistical analyses were conducted with MedCalc 14.8.1 software. Results: A total of eight studies (352 lesions) reporting noninvasive imaging for NMSC clearance following nonsurgical treatment were included. Most (*n* = 7) reported basal cell carcinoma (BCC), and one study reported squamous cell carcinoma (SCC) clearance. A meta-analysis of the BCC clearance revealed that the summary effect for RCM was higher, as compared to the other techniques. Interestingly, the sensitivity and specificity for OCT were 86.4% (95% CI: 65.1–97.1) and 100% (95% CI: 94.8–100.0), respectively, whilst, for RCM, they reached 100% (95%CI: 86.8–100) and 72.5% (95% CI: 64.4–79.7), respectively. Conclusions: Routine clinical examination and dermoscopy underperform when employed for NMSC clearance monitoring, although they represent the first approach to the patient. OCT and RCM seem to improve the detection of persistent BCC after medical treatment.

## 1. Introduction

Non-melanoma skin cancers (NMSC) represent one-third of all malignant tumors [1]. The most common NMSCs are basal cell carcinoma (BCC) and squamous cell carcinoma (SCC) [2]. Treatment is mandatory, although the risk of metastasis is inferior to melanoma. NMSC progression can lead to invasion and the potential damage of nearby tissue. Currently, surgery is widely used [3,4,5], with clearance rates of around 95% [6]. However, in selected NMSCs cases, nonsurgical treatment modalities can be employed as definitive or adjunct therapies, such as in cases of patient surgical ineligibility, multiple lesions, advanced disease and aesthetic site location [7,8,9,10].

Nonsurgical approaches include both topical and systemic medical treatments. Topical treatments, such as immunomodulatory agents, photodynamic therapy (PDT), curettage and cryotherapy, can reach tumor clearance rates of 70–90% for low-risk lesions [11,12]. However, patients with locally advanced or metastatic disease should be considered for systemic therapies [6]. In patients undergoing systemic therapy, the response rate has been estimated to be about 50% [1].

Due to the possible persistence of the skin tumor after treatment, monitoring is necessary to assess the tumor clearance. The development of noninvasive skin imaging techniques, such as dermoscopy, reflectance confocal microscopy (RCM) and optical coherence tomography (OCT), has enabled the early recognition of skin cancer features correlating with histopathology [13,14] and are currently used in treatment monitoring for all types of skin cancer [15]. Dermoscopy is routinely applied in daily practice to visualize microstructures of the skin. RCM and OCT are more advanced imaging tools but are currently available in referral centers only. RCM enables the visualization of cytologic and architectural structures of the epidermis, dermo–epidermal junction (DEJ) and upper dermis in horizontal scans, whilst OCT enables the evaluation of structural, morphologic and vascular details of both horizontal and vertical skin sections [16].

The major RCM criteria for BCC diagnosis are basaloid tumor islands. They usually appear as tightly packed cells surrounded by clefting and can be organized as cords, hyper- or hypo-refractile (dark silhouettes surrounded by highly reflective collagen) islands [15,17,18]. Additional RCM criteria include inflammatory cells or increased vasculature at the dermal level [15]. At OCT, the basaloid island corresponds to ovoid structures located either in the epidermis or dermis [19].

SCC are typified by atypical honeycombed patterns in dermoscopy, which correspond to atypical keratinocytes, dark areas with sharp and irregular contours in RCM, which correspond to ulceration, and irregular dilated vessels in RCM [20,21,22]. SCC is characterized in OCT by a thickened epidermal layer, which may hamper the visualization of the underlying structures, and a loss of clear demarcation of the DEJ, which is considered an indication of invasive disease [23,24].

Although histopathology remains the gold standard for NMSC diagnosis, noninvasive skin imaging has been proven to reduce unnecessary skin biopsies [25] and discover the optimal skin site for a biopsy [26].

A review performed in 2012 reported the identification of tumor persistence with dermoscopy among NMSC following medical treatment [27]. However, the proportion of detection of persisting tumors with RCM and OCT compared to histopathology have been reported in limited studies only. We provide a review and meta-analysis for previously noninvasive imaging techniques for the proportions and diagnostic accuracy of NMSC persistence following medical therapy.

## 2. Materials and Methods

### 2.1. Study Selection Criteria

The study followed the Preferred Reporting Items for Systematic Reviews and Meta-analyses (PRISMA) guidelines [28], according to the Declaration of Helsinki. Study criteria specified the following inclusion criteria:BCC or SCC lesions;medical therapy;monitoring with clinical examination, dermoscopy, RCM and/or OCT, with histopathologic or cyto-diagnostic confirmation of all suspected cases of primary NMSC persistence with noninvasive methodology.

Studies were excluded based on the following criteria:languages other than English;case reports and small case series;<1 month follow-up;use of off-label treatments;lesions with pre-tumoral NMSC diagnoses (actinic keratoses);studies or subset of lesions with an unclear number of persistent NMSC cases.

### 2.2. Data Search

We searched the MEDLINE (PubMed) and Cochrane library databases. The following keywords were used: (“dermoscopy” OR “reflectance confocal microscopy” OR “optical coherence tomography”) AND “treatment” AND (“basal cell carcinoma” OR “squamous cell carcinoma” OR “non melanoma skin cancer”). Truncations and synonyms were included in the search operations. Our search included studies from inception to November 2021.

### 2.3. Study Selection and Data Collection

Two authors independently reviewed the abstracts for study inclusion and exclusion criteria (SG and AA). From each of the included studies, the following data were extracted: first author, year of publication, study design, number of patients, diagnostic instrument type, type of skin cancer, number of lesions, number of persistent cases detected and persistent cases identified by noninvasive monitoring and later confirmed with histopathology/cyto-diagnostic testing.

### 2.4. Statistical Analysis

A proportional meta-analysis was performed with MedCalc (MedCalc Software bvba, version 14.8.1, Ostend, Belgium). The weighted summary proportion under the fixed and random effects model was calculated with Freeman–Tukey’s transformation. Proportions (expressed as percentages ±95% CI) of individual studies were included in the meta-analysis.

The pooled proportion for the fixed effects and the random effects models was performed. The fixed effects model pooled proportion assumed a common true effect among studies: a summary effect is an estimate of the common effect size. The random effects model pooled proportion assumed a variation between studies: a summary effect is the weighted average of the effects.

The pooled proportions (±95% CI) and the overall effect (±95% CI) were assessed with a forest plot graph. The size of the marker is relative to the weights assigned to the different studies. Pooled effects are displayed as diamonds, where its location represents the estimated effect size, and its precision is reflected by its width.

Heterogeneity between studies was assessed by the χ^2^-based Cochran’s Q statistic test and I^2^ metric. Heterogeneity was considered significant at *p* < 0.01 for the Q statistic (if the observed variance exceeds the expected variance), and I^2^ metric (I^2^ = 100% x (Q-df)/Q) uses cut-off points of no heterogeneity (I^2^ = 0–25%), moderate (I^2^ = 25–50%), large (I^2^ = 50–75%) and extreme (I^2^ = 75–100%) heterogeneity. 

Diagnostic accuracy was calculated for lesions considered suspicious for persistent disease with OCT and RCM compared to lesions’ histological diagnostic confirmation. In order to compare the diagnostic accuracy of RCM and OCT to the gold standard method (histology), we estimated the sensitivity and specificity.

## 3. Results

### 3.1. Study Selection

A total of 1025 records were identified with the initial search. Following the removal of duplicates and studies excluded based on the title and abstract assessment, 57 studies were available for full-text assessment. A total of eight studies met the final inclusion eligibility and were included in the qualitative synthesis, and seven were included in the quantitative synthesis (Figure 1).

### 3.2. Study Population

Eight studies, including 200 patients, aged 32–93, with 352 lesions, reported BCC and SCC treatment monitoring. Most studies (*n* = 6) were conducted prospectively, and most lesions were located on the truck (42%) or head and neck (26%). Clinical examination was performed in all studies. Skin imaging devices used for treatment monitoring included dermoscopy alone (*n* = 2), RCM alone (*n* = 1), RCM and dermoscopy (*n* = 3), OCT alone (*n* = 2) and OCT and dermoscopy (*n* = 1). The main characteristics of the selected studies are reported in Table 1.

Seven studies, including 177 patients (223 lesions), reported the proportion of BCC persistence suspected with noninvasive imaging and lesion diagnostic confirmation with histopathology/cyto-diagnostics. The presence or absence of persistent disease was based upon the observation of pre-defined features with clinical examination and noninvasive imaging tools. A meta-analysis for pooled effects was performed for all studies reporting BCC lesions, see Table 2, Figure 2.

Overall, the pooled proportion (95% CI) rate of BCC persistence detected with histopathology was 28.3% (17.9–39.9). The heterogeneity between the studies was assessed as significantly large (Q = 20.8, I^2^ = 71.2%, *p* = 0.002), Figure 3A.

According to the skin imaging device employed for treatment monitoring, the outcomes varied. The rates of residual tumors detected through clinical evaluation and dermoscopy imaging were 20.1% (12–29.7, with significant large heterogeneity Q = 16.7, I^2^ = 64%, *p* = 0.01) and 18.4% (7.8–32.3, with significant extreme heterogeneity Q = 21, I^2^ = 81%, *p* = 0.0003) respectively (see Figure 3B,C). A large prospective study, including 56 patients (98 BCCs) assessed with dermoscopy imaging, highlighted the importance of specific criteria for the identification of persistent disease, including pigmented structures, ulceration/small erosion and arborizing vessels (Figure 4A–E). The authors also suggested that white–red structureless areas and/or superficial fine telangiectasias should be interpreted with caution, as they may also represent post-treatment scarring [30].

The pooled proportion of residual tumors detected with OCT was 20.9% (8.7–36.9). OCT was adopted in only three studies (91 BCCs), and the results rendered a large heterogeneity (Q = 4.7, I^2^ = 57.3%, *p* = 0.096) (see Figure 3D). The OCT features observed in association with the disease clearance included the absence of hyper- or hypo-reflective structures, protruding from the epidermis or surrounded by white stroma (corresponding to BCC tumor lobules observed with histopathology), thickening of the epidermis and the disruption of normal layered skin architecture [31,32,34] (see Figure 4F,G). However, the authors noted that the presence of ‘scar-like’ structures or ulcers may limit the identification of tumor features with OCT [19].

All lesions considered suspicious for tumor persistence with RCM were confirmed with histopathological diagnoses. The pooled proportion of persistent BCCs was estimated to be 32.9% (15.3–53.6). The results were calculated based on four studies reporting 28 persistent lesions, and the heterogeneity was assessed as significantly large (Q = 6, I^2^ = 66.5%, *p* = 0.05) (Figure 3E). The authors underlined that the absence of tumor cords or islands or dark silhouette RCM features was associated with disease clearance [15,29] (see Figure 4H,I). Additionally, the major and minor criteria for a BCC diagnosis with RCM were clearly described by Longo et al. [15].

### 3.3. Diagnostic Accuracy

Among the 91 BCCs assessed with OCT, 22 (24.7%) were identified as suspicious for disease persistence. Of those suspected at OCT, 19 were confirmed with histology, revealing an overdiagnosis for OCT of 15.7% (three false positives). These three false-positive cases were diagnosed at histopathology as one sebaceous hyperplasia and two hyperkeratotic lesions. The OCT sensitivity and specificity estimated were 86.4% (95% CI: 65.1–97.1) and 100% (95% CI: 94.8–100.0), respectively.

Of the 70 BCCs assessed with RCM, 26 (37.1%) were identified as suspicious of disease persistence. A subsequent histopathological/cyto-diagnostic assessment confirmed all RCM suspicious lesions and an additional lesion, not considered as persistent disease by RCM (one false negative), revealing a 96.3% diagnostic accuracy for RCM. Additionally, the RCM sensitivity and specificity were 100% (95% CI: 86.8–100) and 72.5% (95% CI: 64.4–79.7), respectively.

### 3.4. SCC Persistence Review

Only one study reported rates of SCC (Bowen disease) persistence among 29 lesions, correlated with vascular structures, as observed with dermoscopy. In detail, 3 months after the final PDT treatment or 5% imiquimod cream, dermoscopy suspected persistence disease in 16 lesions, of which 14 were confirmed at histopathology. Of the remaining 15 lesions, at a median follow-up of 15 months, 1 lesion was suspected to have persistent disease, and histopathology confirmed this case. Therefore, the dermoscopic vascular structures observed with dermoscopy were positively associated with tumor persistence [5].

## 4. Discussion

From our analysis, clearance rates for medical treatments are confirmed to be inferior to those reported for surgical treatments of BCCs and SCCs. Therefore, patients selected for medical treatment should undergo treatment monitoring, and the proportion analysis suggests that RCM is the most accurate noninvasive diagnostic tool.

The analysis of the current literature revealed a prevalence of studies applying noninvasive skin imaging on NMSC treatment monitoring applied to the BCC, while only one study dedicated to SCC persistent disease met the inclusion criteria for this review. This difference might be explained by the fact that low-risk BCCs are ideal candidates for a medical treatment approach, whereas SCC is more commonly treated with surgical procedures. Additionally, the most frequently used techniques for treatment monitoring are clinical evaluation and dermoscopy, while OCT and RCM have been employed in limited studies. Further, the heterogeneity among the studies was significantly large or extreme for all monitoring techniques.

In daily practice, clinical examination and dermoscopy represent the first approach for treatment monitoring [5,30]. However, according to the observation of the dermoscopy patterns, not all persistent cases confirmed by histopathology were suspected with dermoscopy (95% CI do not overlap). However, although dermoscopy represents a simple, rapid and cost-effective method for treatment monitoring, our data suggest that dermoscopy returns a proportion of false-negative lesions [15,29,30,31,33].

The pooled detection rates for OCT encompassed the range of lesions identified by histopathology but returned a proportion of false positives (one sebaceous hyperplasia and two hyperkeratotic lesions) [31]. Sebaceous hyperplasia represents a differential diagnosis of BCC [35], whilst OCT images of hyperkeratotic lesions are difficult to interpret due to the presence of scaling. These data suggest that slightly more lesions are considered dubious for persistent disease by OCT than are necessary. Overall, OCT showed the highest specificity, being 100% (95% CI: 94.8–100.0), and a sensitivity of 86.4% (95% CI: 65.1–97.1). However, recently, a consensus for OCT patterns associated with BCC diagnoses has been reported [19]. Therefore, OCT’s diagnostic precision for NMSC treatment monitoring is expected to improve in the future.

RCM has been used in research, as well in clinical practice, over the last two decades, and the criteria for NMSC detection have been widely described [15,17,18]. Interestingly, the range of lesions identified by RCM overlap with those identified by histopathology, returning a greater agreement with histopathology as compared to the other techniques, reaching a sensitivity of 100% (95% CI: 86.8–100) and a specificity of 72.5% (95% CI: 64.4–79.7) [15,29,33].

This study is limited by the inclusion of a small number of manuscripts reporting small cohorts with different study designs and therapeutic approaches employed. Clinical evaluation and dermoscopy were employed more often than RCM and OCT; therefore, the features of these first techniques are much more consolidated than those of OCT in particular. The data obtained in this study could not be compared to other data, as there are no studies reporting the diagnostic proportional estimates or diagnostic accuracy available in the literature. However, our results underline the overall previously unreported impact of RCM and OCT techniques in BCC treatment monitoring, but the data should be interpreted with caution due to the significantly large to extreme heterogeneity among the included studies.

## 5. Conclusions

While this study confirmed the wide use of clinical examination and dermoscopy in NMSC treatment monitoring, our results suggest that RCM best identifies BCC lesions with persistent disease following medical treatment. However, the OCT is in evolution, and more data are needed to confirm the noninvasive skin imaging diagnostic accuracy. Due to the high sensitivity of the RCM and specificity of OCT, studies reporting an integrated approach with the two techniques are encouraged.

## Figures and Tables

**Figure 1 cancers-14-02836-f001:**
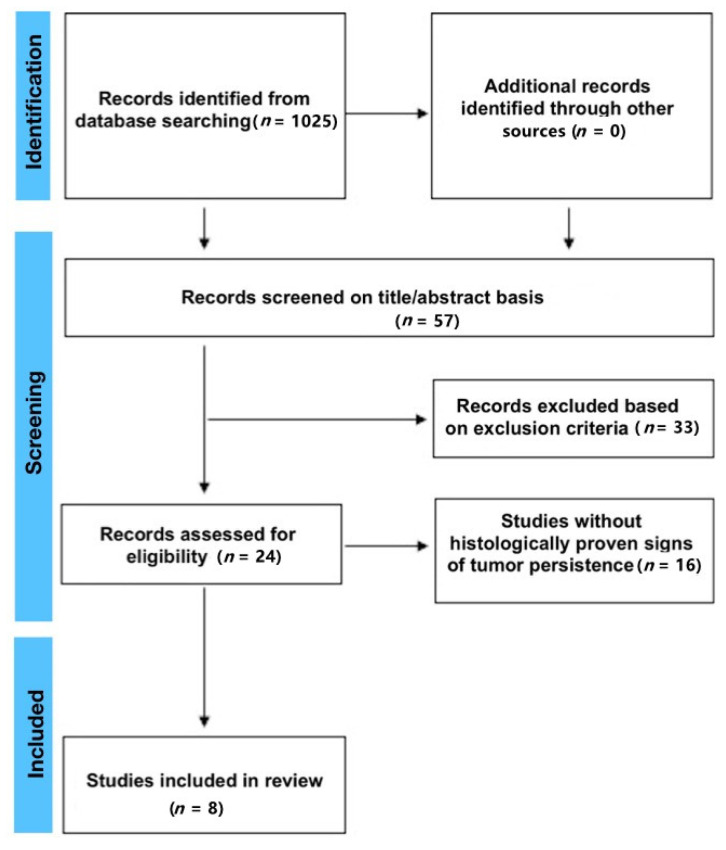
Literature flow diagram for the search and study selections.

**Figure 2 cancers-14-02836-f002:**
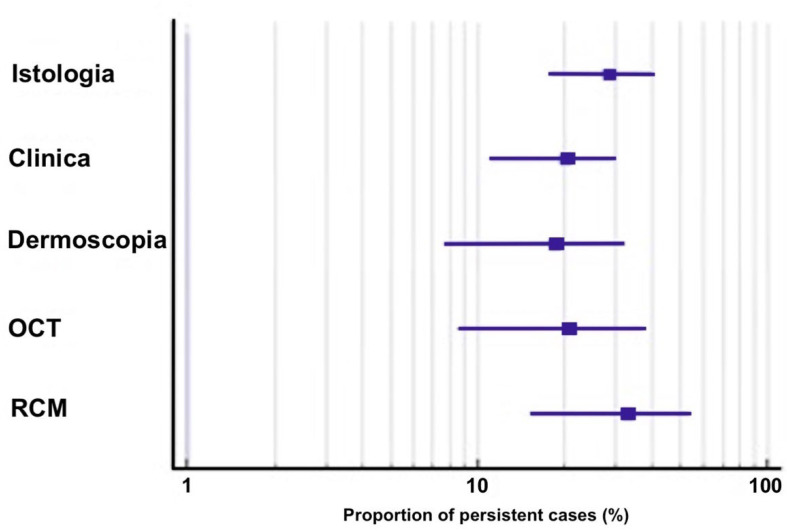
Proportion of the included studies with different detection methods and rates of persistent basal cell carcinomas.

**Figure 3 cancers-14-02836-f003:**
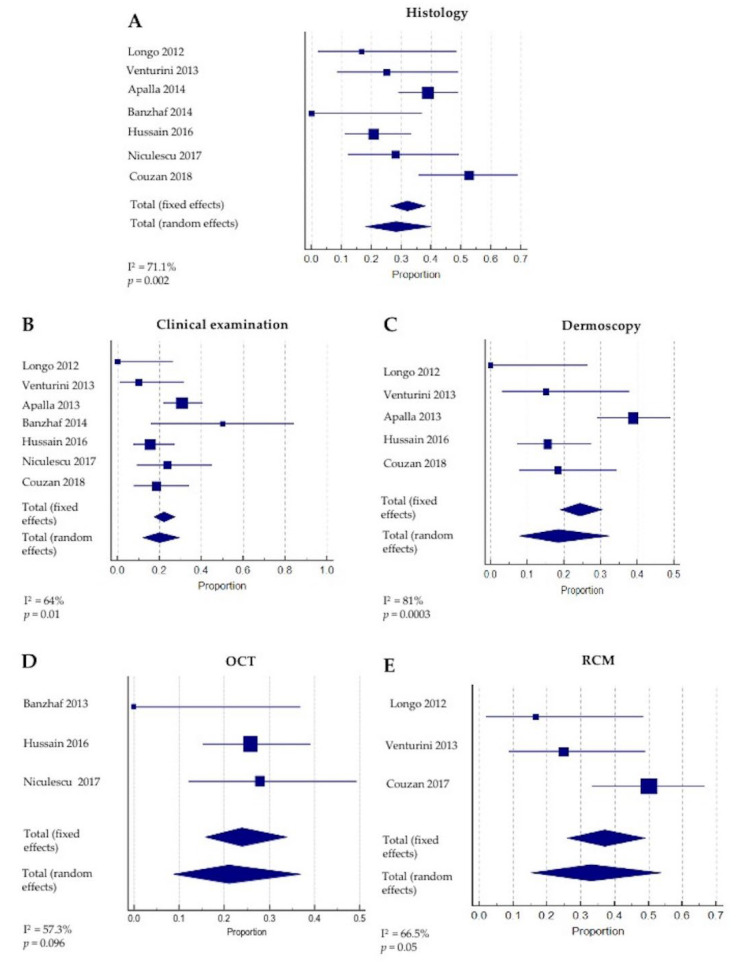
Proportion of included studies and rates of persistent tumors with (**A**) histology, (**B**) clinical examination, (**C**) dermoscopy, (**D**) optical coherence tomography, (OCT) and (**E**) reflectance confocal microscopy (RCM) [15,29,30,31,32,33,34].

**Figure 4 cancers-14-02836-f004:**
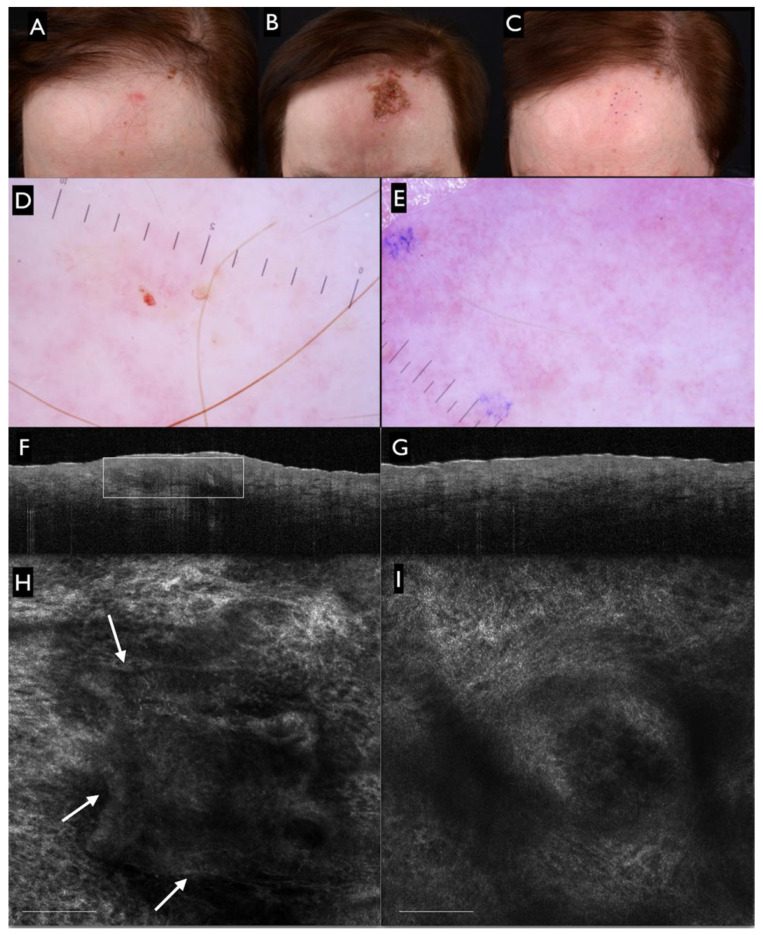
(**A**) Clinical image of a superficial basal cell carcinoma (BCC) from our center archive located on the left forehead of a 60-year-old woman, before initiating imiquimod 5% cream applied 5 times a week for 6 weeks. (**B**) Clinical image two months after the six-week cycle of treatment showing erythema, crusting and erosion. (**C**) Complete clinical healing of the lesion at the 18-month follow-up visit was observed. (**D**) Dermoscopic image of the superficial basal cell carcinoma (BCC) showing small erosions at the baseline and (**E**) after imiquimod treatment. Dermoscopic clearance was observed at the 18-month follow-up. (**F**) Corresponding OCT image shows ovoid structures protruding from the epidermis (white rectangular area) at the baseline and (**G**) complete OCT clearance at the 18-month follow-up (image size 6 mm × 2 mm). (**H**) RCM image shows tumor cords (white arrows) of the BCC at the baseline (**I**), which were no longer observed at the 18-month follow-up (scale bar 100 µm), confirming the RCM clearance.

**Table 1 cancers-14-02836-t001:** Main characteristics of eligible studies.

Included Studies, Author, Year of Publication	Study Type	Lesion Type	Patients, *n*	Lesions, *n*	Lesion Location, *n* (% of Lesions)				Type of Treatment	Follow-Up, Months	Noninvasive Skin Imaging Tools
					Head and neck	Limbs	Trunk	Genitalia			
Longo 2012 [15]	Pro	BCC	10	12	4 (33)	3 (25)	5 (42)	0	PDT	18	Dermoscopy, RCM
Venturini 2013 [29]	Pro	BCC	20	20	6 (30)	2 (10)	12 (60)	0	PDT	3	Dermoscopy, RCM
Apalla 2014 [30]	Pro	BCC	55	98	19 (20)	22 (22)	57 (58)	0	PDT, Imiquimod	3–12	Dermoscopy
Hussain 2016 [31]	Pro	BCC	58	58	n.r.	n.r.	n.r.	n.r.	Curettage ± MAL PDT	1–36	Dermoscopy, OCT
Niculescu 2017 [32]	Retro	BCC	10	25	2 (8)	8 (32)	15 (60)	0	PDT	6	OCT
Couzan 2018 [33]	Retro	BCC	8	38/94 *	45 (48)	24 (25)	25 (27)	0	Vismodegib	18	Dermoscopy, RCM
Banzhaf 2014 [34]	Pro	BCC	16	16	n.r.	n.r.	n.r.	n.r.	Imiquimod	1	OCT
Mun 2012 [5]	Pro	SCC	23	29	0	16 (55)	11 (38)	2 (7)	PDT, Imiquimod	9	Dermoscopy
Total			200	296/352	78 (26)	35 (12)	125 (42)	2 (1)			

* Only 38 lesions were assessed with histopathology and considered for this study. Pro, prospective; Retro, retrospective; BCC, basal cell carcinoma; SCC, squamous cell carcinoma; n.r., not reported; PDT, photodynamic therapy; MAL, Methyl aminolevulinate; RCM, reflectance confocal microscopy; OCT, optical coherence tomography. BCC meta-analysis.

**Table 2 cancers-14-02836-t002:** Individual studies NMSC persistent estimates with histopathology, clinical examination and all noninvasive diagnostic and imaging techniques.

		NMSC Persistence Estimates, *n* (%) 95% Confidence Interval				
Studies	Lesion, n	Histology/Cyto-Diagnostic Confirmation	Clinical Examination	Dermoscopy	OCT	RCM
Longo 2012 [15]	12	2 (16.7) *2.1–48.4*	0 (0.0) *0–26.5*	0 (0) *0–26.5*	NP	2 (16.6) *2.1–48.4*
Venturini 2013 [29]	20	5 (25.0) *8.7–49.1*	2 (10.0) *1.2–31.7*	3 (15) *3.2–37.9*	NP	5 (25) *8.6–49.1*
Apalla 2014 [30]	98	38 (38.8) *29.1–49.1*	30 (30.6) *21.7–40.7*	38 (38.7) *29.1–49.1*	NP	NP
Banzhaf 2014 [34]	8	0 (0.0) *0–36.9*	4 (50.0) *15.7–84.3*	NP	0 (0.0) *0–36.9*	NP
Hussain 2016 [31]	58	12 (20.7) *11.2–33.3*	9 (15.5) *7.3–27.4*	9 (15.5) *7.3–27.4*	15 (25.8) *15.2–39*	NP
Couzan 2018 [33]	38	20 (52.6) *35.8–69*	7 (18.4) *7.7–34.3*	7 (18.4) *7.7–34.3*	NP	19 (50) *33.4–66.6*
Niculescu 2017 [32]	25	7 (28.0) *12.1–49.4*	6 (24.0) *9.3–45.1*	NP	7 (28.0) *12.1–49.4*	NP
Total	**259**	**84 (28.3)** * **17.9–39.9** *	**58 (20.1)** * **12–29.7** *	**57 (18.4)** * **7.7–32.3** *	**22 (20.9)** * **8.7–36.9** *	**26 (32.9)** * **15.3–53.6** *

OCT, optical coherence tomography; RCM, reflectance confocal microscopy; NP non-performed.

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
