# Peer review of "Non-Melanoma Skin Cancer Clearance after Medical Treatment Detected with Noninvasive Skin Imaging: A Systematic Review and Meta-Analysis"

_cancers, 2022, doi:10.3390/cancers14122836_

Round 1

Reviewer 1 Report

the body localization of the lesions should be provided in table 1 or 2. 

Author Response

Report 1: the body localization of the lesions should be provided in table 1 or 2.

Dear Reporter. As per your kind indication, we have added the lesion locations (also in terms of percentages of all lesions enrolled in the study) and added a comment to the main text section regarding the frequencies of the lesion location included in selected studies. Please see the new Table added below:

Included studies, author, year of publication

Study type

Lesion type

Patients, n

Lesions, n

Lesion location, n

(% of lesions)

Type of treatment

Follow-up,

months

Non-invasive skin imaging tools

Head and neck

Limbs

Trunk

Genitalia

Longo, 2012 [15]

Pro

BCC

10

12

4 (33)

3 (25)

5 (42)

0

PDT

18

Dermoscopy, RCM

Venturini, 2013 [29]

Pro

BCC

20

20

6 (30)

2 (10)

12 (60)

0

PDT

3

Dermoscopy, RCM

Apalla, 2014 [30]

Pro

BCC

55

98

19 (20)

22 (22)

57 (58)

0

PDT, Imiquimod

3-12

Dermoscopy

Hussain, 2016 [31]

Pro

BCC

58

58

n.r.

n.r.

n.r.

n.r.

CurettageMAL PDT

1-36

Dermoscopy, OCT

Niculescu, 2017[32]

Retro

BCC

10

25

2 (8)

8 (32)

15 (60)

0

PDT

6

OCT

Couzan, 2018 [33]

Retro

BCC

8

38/94*

45 (48)

24 (25)

25 (27)

0

Vismodegib

18

Dermoscopy, RCM

Banzhaf, 2014 [34]

 Pro

BCC

16

16

n.r.

n.r.

n.r.

n.r.

Imiquimod

1

OCT

Mun, 2012 [5]

Pro

SCC

23

29

0

16 (55)

11 (38)

2 (7)

PDT, Imiquimod

9

Dermoscopy

Total

200

296/352

78 (26)

35 (12)

125 (42)

2 (1)

* only 38 lesions were assessed with histopathology and considered for this study.

Pro, prospective; Retro, retrospective; BCC, basal cell carcinoma; SCC, squamous cell carcinoma; n.r., not reported; PDT, photodynamic therapy; MAL, Methyl aminolevulinate; RCM, reflectance confocal microscopy; OCT, optical coherence tomography.

Reviewer 2 Report

In this manuscript, it was aimed to review non-invasive imaging techniques using in detection of persistent basal cell carcinoma and squamous cell carcinoma after medical treatment. In order to estimate the diagnostic accuracy of optical coherence tomography and reflectance confocal microscopy, the authors carried out meta-analysis.

The manuscript is well focused and clear in terms of text and the conclusions are supported by the data. However, there are some minor issues to be clarified by the authors. In conclusion, this manuscript should be subjected to a minor revision to be deemed for publication in Cancers.

  1. In the Abstract, in line 41. the abbreviation SCC should be introduced.
  2. The Introduction should be extended with the already known RCM and OCT features of BCC and SCC.
  3. All included study should be supported by references. Please insert the appropriate references into Table 1. after the author and year.
  4. Line 192 is not supported by reference No. 22 and No. 23. These articles do not discuss OCT features.
  5. In the Discussion section many sentences are not supported by references. (e.g: “The pooled detection rates for OCT encompassed the range of lesions identified by histopathology, but returned a proportion of false positives (1 sebaceaous hyperplasia and 2 hyperkeratotic lesions)” or “RCM has been used in research as well in clinical practice over the last two decades and criteria for NMSC detection have been widely described.”
  6. From the last sentence of the last paragraph of the Introduction “We provide a review and meta-analysis for previously unreported non-invasive imaging techniques for the proportions and diagnostic accuracy of NMSC persistence following medical therapy.” the word “unreported” should be deleted, as here previously reported findings are reviewed.
  7. Prior to publication, the grammar of the manuscript should be revised e.g. page 2, line 72. the word “tolls” should be corrected (“tools”; page 8, line 213 “Dermoscoopic” (“Dermoscopic”); page 9, line 261 “sebaceaous” (“sebaceous.”)

  8. From the Author contributions section the original instruction should be removed „For research articles with several authors, a short paragraph specifying their 291 individual contributions must be provided. The following statements should be used “; ” Please turn to the CRediT taxonomy for the term explanation. Authorship must be limited to those who have contributed substantially to the work reported.”
  9. The Ackowledgments section should be listed or deleted.
  10. From the Confict of interest statement, apostrophes should be deleted.

Author Response

In this manuscript, it was aimed to review non-invasive imaging techniques using in detection of persistent basal cell carcinoma and squamous cell carcinoma after medical treatment. In order to estimate the diagnostic accuracy of optical coherence tomography and reflectance confocal microscopy, the authors carried out meta-analysis.
The manuscript is well focused and clear in terms of text and the conclusions are supported by the data. However, there are some minor issues to be clarified by the authors. In conclusion, this manuscript should be subjected to a minor revision to be deemed for publication in Cancers.

R: thank you for taking the time to review our article and for the appreciation and suggestions to improve it.

In the Abstract, in line 41. the abbreviation SCC should be introduced.

R: it has been introduced, as suggested

The Introduction should be extended with the already known RCM and OCT features of BCC and SCC.

R: the introduction has been extended, following reviewer’s suggestion

All included study should be supported by references. Please insert the appropriate references into Table 1. after the author and year.

R: appropriate references have been added

Line 192 is not supported by reference No. 22 and No. 23. These articles do not discuss OCT features.

R: appropriate references have been added

In the Discussion section many sentences are not supported by references. (e.g: “The pooled detection rates for OCT encompassed the range of lesions identified by histopathology, but returned a proportion of false positives (1 sebaceaous hyperplasia and 2 hyperkeratotic lesions)” or “RCM has been used in research as well in clinical practice over the last two decades and criteria for NMSC detection have been widely described.”

R: appropriate references have been added

From the last sentence of the last paragraph of the Introduction “We provide a review and meta-analysis for previously unreported non-invasive imaging techniques for the proportions and diagnostic accuracy of NMSC persistence following medical therapy.” the word “unreported” should be deleted, as here previously reported findings are reviewed.

R: it has been deleted, as suggested

Prior to publication, the grammar of the manuscript should be revised e.g. page 2, line 72. the word “tolls” should be corrected (“tools”; page 8, line 213 “Dermoscoopic” (“Dermoscopic”); page 9, line 261 “sebaceaous” (“sebaceous.”)

R: it has been checked and corrected

From the Author contributions section the original instruction should be removed „For research articles with several authors, a short paragraph specifying their 291 individual contributions must be provided. The following statements should be used “; ” Please turn to the CRediT taxonomy for the term explanation. Authorship must be limited to those who have contributed substantially to the work reported.”
The Ackowledgments section should be listed or deleted.
From the Confict of interest statement, apostrophes should be deleted.

R: the mentioned sections have been edited according to reviewer’s suggestions

Reviewer 3 Report

I have read the paper with interest.

It is not clear if FU skin biopsies were performed in all cases. If not, it would be hard to assess the real sensitivity and specificity of the non-invasive skin imaging in detecting remnant lesions. There should be clarification regarding this.

Author Response

It is not clear if FU skin biopsies were performed in all cases. If not, it would be hard to assess the real sensitivity and specificity of the non-invasive skin imaging in detecting remnant lesions. There should be clarification regarding this.

R: Thank you for your comment.

We modified this sentence in the manuscript “Study criteria specified the inclusion of BCC or SCC lesions treated with medical therapy and monitored with clinical examination, dermoscopy, RCM and/or OCT, with histopathologic or cytodiagnostic confirmation of “ALL” suspected cases of primary NMSC persistence with non-invasive methodology” in order to clarify that all suspected cases had histopathologic or cytodiagnostic confirmation.

Reviewer 4 Report

The authors investigated non-melanoma skin cancer clearance after medical treatment detected with non-invasive skin imaging.

The study is limited by the inclusion of a small number of manuscripts as pointed out by the authors. Therefore, it would make sense to include also surgically removed skin tumours. Furthermore, some of the included studies did not confirm the imaging diagnosis by histopathology, which makes the studies of dubious quality.

Location of the included tumours should be mentioned. Is the skin thick or thin like the eyelids.

 Study inclusions should be presented using bullets (similar to exclusion).

Author Response

The authors investigated non-melanoma skin cancer clearance after medical treatment detected with non-invasive skin imaging.

The study is limited by the inclusion of a small number of manuscripts as pointed out by the authors. Therefore, it would make sense to include also surgically removed skin tumours.

This is an interesting suggestion, however whilst assesssing the literature, we noticed that there would have only been one manuscript meeting our inclusion criteria, who reported a surgical approached for NMSC with treatment monitoring (Navarrete-Dechent C, et al. Management of complex head-and-neck basal cell carcinomas using a combined reflectance confocal microscopy/optical coherence tomography: a descriptive study. Arch Dermatol Res. 2021 Apr;313(3):193-200. doi: 10.1007/s00403-020-02037-6).

However, this study included evaluations performed at bedside and did not include any follow up.

Furthermore, some of the included studies did not confirm the imaging diagnosis by histopathology, which makes the studies of dubious quality.

The imaging diagnosis of suspected persistent cases included in this metanalysis were all confirmed with histopathology/cytodiagnostic exam. For the only study with partial lesion confiormation (Cousan) only the lesions with histopathology confirmation were considered and included in our metanalaysis. This has been specified in the Table 1. legend which reads, “* only 38 lesions were assessed with histopathology and considered for this study.”

Location of the included tumours should be mentioned. Is the skin thick or thin like the eyelids.

Thank you.  We have added the lesion location for each study into Table 1. Many thanks for your suggestion.

Study inclusions should be presented using bullets (similar to exclusion).

Many thanks.  We have modified the study inclusion criteria as requested.

Round 2

Reviewer 3 Report

The revised version of the paper is acceptable for publication.